# Olprinone, a Selective Phosphodiesterase III Inhibitor, Has Protective Effects in a Septic Rat Model after Partial Hepatectomy and Primary Rat Hepatocyte

**DOI:** 10.3390/ijms25137189

**Published:** 2024-06-29

**Authors:** Masaya Kotsuka, Tetsuya Okuyama, Yuki Hashimoto, Hiroaki Kitade, Mikio Nishizawa, Katsuhiko Yoshizawa, Richi Nakatake

**Affiliations:** 1Department of Surgery, Kansai Medical University, Hirakata 573-1010, Japan; kotsukam@hirakata.kmu.ac.jp (M.K.); okuyamat@hirakata.kmu.ac.jp (T.O.); hashimoy@hirakata.kmu.ac.jp (Y.H.); kitadeh@takii.kmu.ac.jp (H.K.); 2Department of Biomedical Sciences, College of Life Sciences, Ritsumeikan University, 1-1-1 Nojihigashi, Kusatsu 525-8577, Japan; nishizaw@sk.ritsumei.ac.jp; 3Department of Innovative Food Sciences, School of Food Sciences and Nutrition, Mukogawa Women’s University, 6-46 Ikebiraki-cho, Nishinomiya 663-8558, Japan; yoshizak@mukogawa-u.ac.jp

**Keywords:** olprinone, partial hepatectomy with lipopolysaccharide, primary cultured hepatocytes, acute liver injury, inducible nitric oxide synthase, nuclear factor kappa B

## Abstract

Olprinone (OLP) is a selective inhibitor of phosphodiesterase III and is used clinically in patients with heart failure and those undergoing cardiac surgery; however, little is known about the effects of OLP on hepatoprotection. The purpose of this study aimed to determine whether OLP has protective effects in in vivo and in vitro rat models of endotoxin-induced liver injury after hepatectomy and to clarify the mechanisms of action of OLP. In the in vivo model, rats underwent 70% partial hepatectomy and lipopolysaccharide treatment (PH/LPS). OLP administration increased survival by 85.7% and decreased tumor necrosis factor-α, C-X-C motif chemokine ligand 1, and inducible nitric oxide synthase (iNOS) mRNA expression in the livers of rats treated with PH/LPS. OLP also suppressed nuclear translocation and/or DNA binding ability of nuclear factor kappa B (NF-κB). Pathological liver damage induced by PH/LPS was alleviated and neutrophil infiltration was reduced by OLP. Primary cultured rat hepatocytes treated with the pro-inflammatory cytokine interleukin-1β (IL-1β) were used as a model of in vitro liver injury. Co-treatment with OLP inhibited dose-dependently IL-1β-stimulated iNOS induction and NF-κB activation. Our results demonstrate that OLP may partially inhibit the induction of several inflammatory mediators through the suppression of NF-κB and thus prevent liver injury induced by endotoxin after liver resection.

## 1. Introduction

Resection of the liver is the established standard of care for the treatment of patients with intrahepatic neoplasms. Liver resection has a mortality rate of less than 5% in specialized centers [1]. Improved safety due to advances in instrumentation and increased anatomic understanding of the liver have expanded the indications for liver resection and the range of liver volumes to be resected. When liver volume is reduced through surgery, patients are more susceptible to subsequent infections and sepsis [2]. The loss of functional liver parenchyma due to cirrhosis can result in impaired liver function and an inability to regenerate the liver, leading to liver failure [3]. These observations in clinical practice are corroborated by experimental data showing that the sensitivity to LPS is increased in animals that have undergone hepatectomy [4,5]. Countermeasures against sepsis caused by postoperative infection after hepatic resection are important challenges in hepatic surgery. 

Olprinone (OLP) is one of the selective inhibitors of phosphodiesterase III (PDE-III), which increases intracellular cyclic adenosine monophosphate (cAMP) levels by inhibiting cAMP degradation in vascular smooth muscle cells and cardiomyocytes. Increased intracellular cAMP levels in these cells lead to both inotropic and vasodilator effects [6]. In clinical practice, OLP is used during cardiac surgery and in patients with heart failure. Previous investigations have shown the beneficial effects of PDE-III inhibitors, including improving microcirculation and reducing inflammation [7,8,9]. As PDE-III is abundant in the liver [10,11] and OLP induces antioxidant stress responses in the liver [12], OLP may also have hepatoprotective effects. However, few studies have examined whether OLP treatment affects liver injury in animals after hepatectomy. Based on the above, we hypothesized that pretreatment with OLP may prolong survival by reducing susceptibility to endotoxic liver injury induced by hepatectomy. An experimental rat model in which lipopolysaccharide (LPS) is administered after approximately 70% hepatectomy was used to assess this hypothesis. We examined the hepatoprotective effects of OLP and its possible mechanisms using primary cultures of rat hepatocytes as an in vitro model of liver injury [13].

## 2. Results

### 2.1. OLP Improves Survival in 70% Partial Hepatectomy and LPS (PH/LPS)-Treated Rats

To produce experimental models of severe acute liver injury in animals, frequently 70% partial hepatectomy (PH) is combined with LPS, resulting in mortality rates of 60–100%. In this study, rats in the liver injury model receiving OLP at a dose of 10 mg/kg body weight (Figure 1A, open circles) had significantly increased survival (85.7%) compared to rats treated with PH/LPS alone (black circles). The PH/LPS + lower OLP (3.3 mg/kg) group (grey closed circles) did not increase survival (57.1%) as much as the PH/LPS + OLP (10 mg/kg) group and was not significantly different from the PH/LPS-only group. Rats receiving OLP above a dose of 10 mg/kg exhibited adverse effects. Treatment with OLP did not affect serum aspartate transaminase (AST) or alanine transaminase (ALT) levels in PH/LPS rats (Figure 1B,C).

### 2.2. OLP Decreases Pro-Inflammatory Mediator Expression in PH/LPS-Treated Rats

Although the mRNA expression of hepatic pro-inflammatory genes in rat livers increased after treatment with PH/LPS, the PH/LPS + OLP group had lower tumor necrosis factor-α (TNF-α, 1 and 4 h), interleukin (IL)-6 (1 h), and C-X-C motif chemokine ligand 1 (CXCL1, 1 and 4 h) mRNA expression (Figure 2A–C). Serum analysis revealed that OLP decreased the pro-inflammatory cytokine levels, including TNF-α (1 h), IL-6 (4 h), and IL-1β (1 and 4 h), in PH/LPS-treated rats (Figure 2D–F).

### 2.3. OLP Reduces Inducible Nitric Oxide Synthase (iNOS) mRNA Induction and Nitric Oxide (NO) Production by Suppressing Nuclear Factor Kappa B (NF-κB) Activation in PH/LPS-Treated Rats

PH/LPS treatment increased the hepatic iNOS mRNA expression, whereas OLP treatment inhibited its expression at 4 h (Figure 3A). Because serum NO levels are more pronounced after 3 h of PH/LPS treatment [14], NO levels were measured at 4 h in this study and showed a significant increase; however, OLP treatment inhibited this increase (Figure 3B). Electrophoretic mobility shift assay (EMSA) showed that OLP suppressed nuclear translocation or DNA binding of NF-κB by PH/LPS at 1 h (Figure 3C).

### 2.4. OLP Decreases Pathological Liver Damage in PH/LPS-Treated Rats

The rat livers were histologically examined 4 h after PH/LPS treatment. Liver sections were stained with hematoxylin and eosin (H&E) and representative images are shown in Figure 4A. PH/LPS treatment induces pathological liver damage, including inflammatory cell infiltration and hemorrhagic manifestations. Single-cell and focal necrosis with ballooning degeneration was also observed (Figure 4A; middle panel). However, OLP treatment decreased the extent and incidence of these pathological changes (Figure 4A right panel and B, Appendix A). Neutrophil infiltration was detected by immunostaining for MPO (Figure 4C). PH/LPS markedly increased neutrophil infiltration (Figure 4C, middle panel). OLP treatment significantly decreased the number of MPO-positive neutrophils in the liver (Figure 4C right panel and D, Appendix A). Representative images are shown in Figure 4E after the terminal deoxynucleotidyl transferase-mediated deoxyuridine triphosphate-digoxigenin nick-end labeling (TUNEL) staining. PH/LPS increased hepatocyte apoptosis, whereas no significant difference in TUNEL-positive cells was observed between the two groups (Figure 4F, Appendix A).

### 2.5. OLP Inhibits iNOS Gene Induction and NO Production, Suppressing NF-κB Activation in Cultured Hepatocytes

Primary rat hepatocyte cultures induced iNOS protein expression and NO production upon IL-1β stimulation, whereas the co-addition of OLP and IL-1β dose-dependently reduced NO production (Figure 5A). NO production was reduced by more than 80% at a dose of 1 mM (250 μg/mL) of OLP. The highest dose of OLP reduced iNOS protein (Figure 5B) and mRNA expression (Figure 5C). Furthermore, OLP inhibited the activation of NF-κB (Figure 5D), a key signaling pathway for iNOS induction. LDH activity in the medium, indicating cytotoxicity, was low after the hepatocytes were treated with OLP (<5% of the whole cell extract activity).

### 2.6. OLP Decreases Pro-Inflammatory Mediator Expression in Cultured Hepatocytes

Real-time PCR analysis showed that OLP decreased mRNA expression of pro-inflammatory cytokine (TNF-α, IL-6, and IL-1β) and chemokine (CXCL1) (Figure 6), similarly to results in the in vivo study.

## 3. Discussion

In the present study, we investigated the effects of OLP on hepatoprotection in PH/LPS-treated rats. The results obtained in the in vitro liver injury model using IL-1β-stimulated rat hepatocytes suggested possible mechanisms of action. In endotoxemia after hepatectomy, liver failure is considered a trigger for progression to multiple organ failure (MOF). OLP reduced pathological liver damage (Figure 4A) and neutrophil infiltration (Figure 4B) in the livers of PH/LPS-treated rats, resulting in markedly improved rat survival (Figure 1A).

In endotoxemia, Kupffer cells play an important role in immune response [15]. At the earliest stage, LPS stimulates Kupffer cells via toll-like receptor 4 (TLR4) [16]. Subsequently, activated Kupffer cells produce inflammatory mediators, including pro-inflammatory cytokines and chemokines, which recruit neutrophils to the liver [17,18]. The inflammatory cytokines produced stimulate Kupffer cells and hepatic parenchymal cells to induce NF-κB activation. When activated, nuclear translocation of NF-κB is induced, thereby enabling DNA binding, activating transcription and leading to overproduction of inflammatory mediators. In the present study, the EMSA experiments revealed that OLP inhibited NF-κB activation in the liver (Figure 3C). Thus, OLP suppressed hepatic TNF-α, IL-6, and CXCL1 mRNA expression (Figure 2A–C). The serum levels of pro-inflammatory cytokines, such as TNF-α, IL-6, and IL-1β, were also decreased by OLP (Figure 2D–F), suggesting that OLP suppressed the transcription of these inflammatory mediators by NF-κB. Additionally, iNOS is induced by NF-κB and exerts a significant impact on liver damage in endotoxemia through the excessive production of NO. The upregulation of iNOS and pro-inflammatory cytokines in inflamed hepatocytes is central to liver injury. In cultured hepatocytes, OLP inhibited NO production and iNOS induction, in part, by blocking the activation of NF-κB (Figure 5). CXCL1 plays a pivotal role in neutrophil mobilization in acute and chronic liver failure, and the suppression of CXCL1 attenuates neutrophil infiltration and decreases apoptosis of hepatocytes by reducing ROS levels [19]. In contrast, pathological TUNEL assays indicated no significant differences in TUNEL-positive cells between the PH/LPS-only and PH/LPS + OLP groups (Figure 4C). The progression of hepatocyte apoptosis may be insufficient to detect apoptotic cells using TUNEL assay. However, delaying sample collection was difficult because of the high probability of mortality. In addition, OLP treatment did not significantly reduce serum ALT and AST (Figure 1B,C); other markers of liver injury, such as the release of miR-122, need to be investigated for therapeutic effects of OLP [20].

OLP has been reported to upregulate hepatic eNOS expression and postoperative eNOS activity, possibly attenuating shear stress after excessive hepatectomy in rats by suppressing cell injury in sinusoidal endothelium and apoptosis of hepatocytes [6]. OLP pretreatment has also been demonstrated to protect sinusoidal endothelial cells from monocrotaline toxicity and prevent the development of sinusoidal obstruction syndrome [21]. OLP has also been reported to protect the liver, intestines, kidneys, and heart against ischemia–reperfusion injury. The anti-inflammation and improvement of microcirculation by increasing cAMP are thought to be the underlying mechanisms that protect the organs of OLP in rats and mice [9,22,23]. In particular, the protective effects of OLP against hepatic ischemia–reperfusion injury have been reported to be explained by increasing the levels of cAMP and suppressing the production of pro-inflammatory cytokines and expression of intercellular adhesion molecule 1 in hepatocytes, possibly by interfering with the signaling pathways of MAPKs and NF-κB [24]. These mechanisms may also be involved in the hepatoprotective effects of OLP against endotoxemia. On the other hand, pro-inflammatory reactions are important for liver regeneration. Investigating the association between anti-inflammatory effects of OLP treatment and liver regeneration in models of milder injury is an important topic for future studies.

The dose and method of OLP administration used in this study (10 mg/kg, single intraperitoneal dose) were different from standard clinical use (2 μg/kg/min, continuous intravenous dose). The dose was calculated based on the previous use in an experimental study [6]; however, unlike the continuous infusion of OLP in previous studies, we administered a single intraperitoneal bolus dose. In applying OLP, attention should be paid to side effects, such as hypotension and tachycardia [25,26]. Because the half-life of OLP is approximately 90 min, a single intraperitoneal dose of OLP was administered 1 h before LPS administration to monitor its effects on hyperthermic shock and side effects. The benefits of continuous infusion of OLP in the PH/LPS model should be considered in future studies. The impact of cumulative/overdosage on therapeutic outcomes is another important issue that should be investigated in the future. The relatively different doses and methods of administration used in the in vivo and in vitro experiments compared to clinical use in humans are limitations of this study. Clinical reports suggest that OLP may protect the liver and spleen after cardiac operations [25]. However, few clinical trials have examined the prevention of liver dysfunction after hepatic resection by OLP. Thus, the hepatoprotective and adverse effects of OLP deduced using these models need to be confirmed in humans.

## 4. Materials and Methods

### 4.1. Materials

The OLP (Coretec) was purchased from Eizai Co., Ltd. (Tokyo, Japan). Recombinant human interleukin 1β (IL-1β) (2 × 10^7^ U/mg protein) was obtained from MyBioSource, Inc. (San Diego, CA, USA). [γ-32P]-ATP was purchased from PerkinElmer Inc. (Waltham, MA, USA). Male Sprague-Dawley (SD) and Wistar rats were purchased from Jackson Laboratory, Inc. (Yokohama, Japan). The rats were kept at 22 °C with a 12 h light/dark cycle for at least 7 days to allow for acclimatization. Standards outlined in the ARRIVE [27] and PREPARE [28] guidelines were followed for animal care and experiments. The animal protocol for this study was approved by the Animal Care Committee of Kansai Medical University (Osaka, Japan) (Approval no. 22-039(1) and no. 22-040(1)).

### 4.2. The Procedure of PH/LPS in Rats

To produce the PH/LPS model, SD rats (240–260 g; 7 weeks old) were anesthetized using isoflurane and a combination of medetomidine, midazolam, and butorphanol before undergoing 70% hepatectomy, as reported previously [4,5,29]. The rats were randomly divided into PH/LPS, PH/LPS + OLP, and control groups. Forty-eight hours after hepatectomy, LPS was administered into the penile vein at a dose of 25 μg/kg body weight (PH/LPS and PH/LPS + OLP groups). The rats assigned to the PH/LPS + OLP group were administered intraperitoneally with one of two OLP doses (3.3 or 10 mg of olprinone hydrochloride hydrate/kg body weight) 1 h before LPS treatment. Survival was monitored for 5 d after LPS injection. Blood and liver samples were obtained from rats 1 and 4 h after LPS administration. The rats were euthanized when they appeared weak and moribund as congestion and multiorgan failure progressed. We used the NIH Office of Animal Care and Use [30] score and severity assessment to assess the animals following liver resection [31].

### 4.3. Culture of Primary Rat Hepatocyte Preparation

To isolate hepatocytes, livers of Wistar rats (200–250 g, 6–7 weeks old) were perfused with collagenase (FUJIFILM Wako Pure Chemical Corp., Osaka, Japan) [32,33]. Williams’ medium E supplemented with 10% fetal calf serum, N-(2-hydroxyethyl)piperazine-N’-2-ethanesulfonic acid (5 mmol/L), antibiotics (streptomycin; 100 μg/mL, penicillin; 100 U/mL, and amphotericin B; 0.25 μg/mL), a protease inhibitor (aprotinin; 0.1 μg/mL, Roche, Basel, Switzerland), and hormones (insulin; 10 nmol/L and dexamethasone; 10 nmol/L) was used suspension for the isolated hepatocytes. After seeding the suspended cells at a density of 1.2 × 10^6^ into 35 mm diameter dishes, the cells were incubated for 2 h and another 3 h at 37 °C in an incubator (5% CO_2_). After incubation, the medium was sequentially replaced with fresh serum-free medium and serum- and hormone-free medium. The cells were further cultured overnight. The stock solution of 3.28 mM OLP was diluted to 0.2–1 mM in fresh serum- and hormone-free medium. After washing with fresh medium at the next day after cell culture, the medium was replaced with that containing OLP. Hepatocytes were then co-incubated with IL-1β (1 nmol/L).

### 4.4. RT-PCR of the In Vivo Rat Model and the Cultured Hepatocytes 

Total RNA was extracted from the cultured hepatocytes or liver samples using Sepasol I Super G (Nacalai Tesque Inc., Kyoto, Japan). From extracted total RNA, cDNA for mRNAs was synthesized using oligo (dT) primers. The primer pairs listed in Appendix A were used and touchdown qPCR was then performed with Rotor-Gene Q (Qiagen, Valencia, CA, USA). The obtained mRNA level values were normalized to value of elongation factor-1α (EF) mRNA. The normalized value for each gene in negative control or IL-1β only treatment was set as 1.0 or 100%, respectively.

### 4.5. Serum Biochemical Analyses of the In Vivo Rat Model

Serum nitrate and nitrite levels (stable NO metabolites) were quantified using a NO colorimetric assay kit (Roche, Mannheim, Germany) based on the Griess method [34]. AST and ALT activities in the serum were quantified by Transaminase C2-Test kit (FUJIFILM Wako Pure Chemical Corp.). Enzyme-linked immunosorbent assays were performed to quantify the serum concentration of pro-inflammatory cytokines were quantified by using commercial kits (Thermo Fisher Scientific Inc., Waltham, MA, USA; R&D Systems, Minneapolis, MN, USA; Proteintech, Rosemont, IL, USA; Arigo Biolaboratories Corp., Hsinchu, Taiwan).

### 4.6. EMSA of the Nuclear Extracts

From liver specimens or cultured hepatocytes, nuclear extracts were prepared. The protein concentration was determined by the Bradford method. After labeling double-stranded DNA probes (sense strand: 5′-AGTTGAGGGACTTTCCCAGGC) that contain a κB site with [γ-32P]-ATP, the probes (40,000 dpm) were incubated with nuclear extracts. Binding to the probes was visualized by autoradiography on the dried gels after resolution by polyacrylamide gel electrophoresis (PAGE) [35]. 

### 4.7. Histopathological Analysis of the In Vivo Rat Model

The liver specimens were fixed in phosphate-buffered saline containing 10% formalin and embedded in paraffin. Sections were cut at 3–5 μm in thickness and stained with H&E. The degree of liver damage was scored based on single-cell necrosis, balloon degeneration, focal necrosis, hemorrhagic manifestations, and inflammatory cell infiltration. Lesion severity was rated on a 4-point scale (0 = no change, 1 = minimal, 2 = mild, and 3 = marked). The total scores were graded for each rat. A toxicologic pathologist certified by the International Federation of Societies of Toxicologic Pathologists (K.Y.) performed the histopathologic evaluation according to predefined histopathologic terminology and diagnostic criteria [36,37]. Neutrophils infiltrated liver tissue were detected by immunostaining for MPO using antibodies (Dako, Carpinteria, CA, USA). Counterstaining was performed with hematoxylin. Nuclear apoptotic bodies in the hepatocytes were detected by TUNEL staining using in situ Apoptosis Detection Kit (MK500; Takara Bio Inc., Kusatsu, Shiga, Japan). Analysts blinded to the treatment counted the number of MPO- and TUNEL-positive cells per square millimeter.

### 4.8. Determination of NO Production and Lactate Dehydrogenase (LDH) Activity in the Cultured Hepatocytes

In the culture medium of hepatocytes, the amount of nitrite, one of the stable metabolites of NO, was measured using the Griess method [34]. In the same medium, cell viability was assessed by measuring the released lactate dehydrogenase (LDH) activity using the Cytotoxicity LDH Assay Kit-WST (Dojindo Laboratories, Kumamoto, Japan).

### 4.9. Western Blot Analyses

Hepatocytes were lysed with sample buffer containing 125 mmol/L Tris–HCl, pH 6.8, 5% glycerol, 2% sodium dodecyl sulfate (SDS), and 1% 2-mercaptoethanol. The extracted proteins were separated with SDS-PAGE and immunostained with primary antibodies (rabbit iNOS; Affinity BioReagents, Golden, CO, USA and mouse β-tubulin; Clone TUB2.1; Sigma-Aldrich Corp., St. Louis, MO, USA; internal control). Immunoreactive proteins were detected by reaction with an enhanced chemiluminescence detection kit (GE Healthcare Biosciences, Piscataway, NJ, USA).

### 4.10. Statistical Analyses

Quantitative data were collected from three to five rats per group or from at least three independent experiments, and the mean values and standard deviations were calculated. Student’s t-test was performed to analyze the differences between two groups. One-way analysis of variance (ANOVA) followed by Tukey–Kramer and Dunnett’s multiple comparison tests were used to analyze the differences between more than three groups. The Kaplan–Meier method was used to construct survival curves and the log-rank test was used for statistical analysis of the effect on survival. Statistical analyses were performed using the JMP 17.2 statistics software (SAS Institute Inc., Cary, NC, USA). *p* values were considered significant at less than 0.05 and less than 0.01 compared with PH/LPS-only rats or IL-1β alone.

## 5. Conclusions

OLP prevented pro-inflammatory mediator production by suppressing NF-κB activation in both in vivo and in vitro liver injury models. Therefore, OLP may have resulted in a significant increase in the survival of the PH/LPS-treated rats. These findings suggest a possible preventive effect of OLP on liver injury.

## Figures and Tables

**Figure 1 ijms-25-07189-f001:**
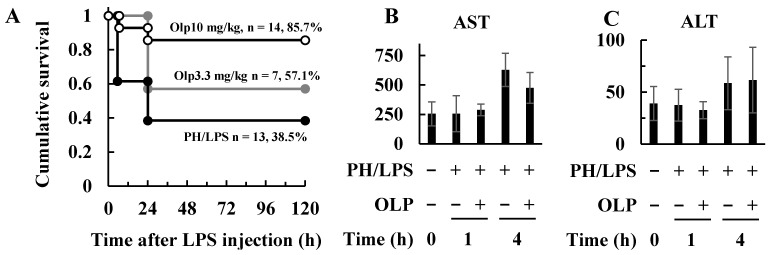
Increased survival with olprinone (OLP) in rat PH/LPS model. (**A**) Effects of OLP on PH/LPS-treated rat survival. OLP (3.3, 10 mg/kg body weight) was administered intraperitoneally 1 h before LPS administration. Cumulative survival was plotted as Kaplan–Meier survival curves for the following groups: PH/LPS-only (positive control; black closed circles), PH/LPS + OLP (3.3 mg/kg; grey closed circles), and PH/LPS + OLP (10 mg/kg; opened circles). The values represent the number of rats and the percentage of survival at 120 h after LPS treatment. Effects of OLP on serum AST (**B**) and ALT (**C**). Bar graph values indicate the mean ± standard deviation (*n* = 3−5 rats per group per time-point).

**Figure 2 ijms-25-07189-f002:**
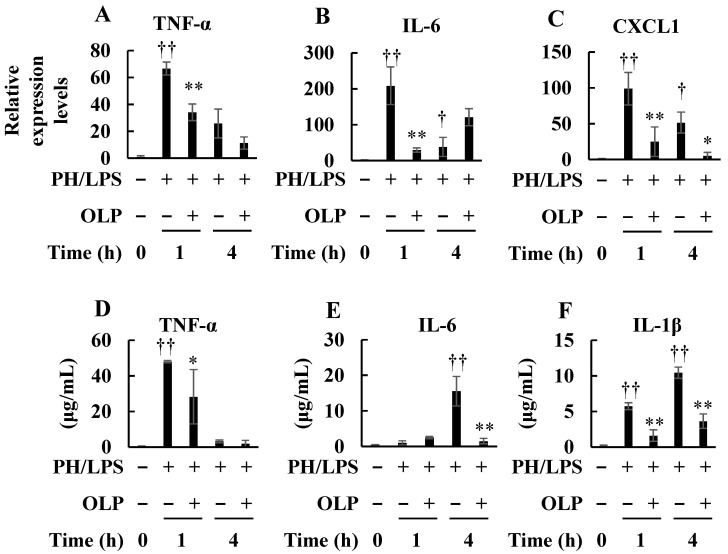
Suppression of induction of pro-inflammatory cytokine and chemokine expression and production by olprinone (OLP). Effects of OLP on mRNA expression and production of pro-inflammatory cytokines and chemokines in PH/LPS-treated rats. (**A**) TNF-α, (**B**) IL-6, and (**C**) CXCL1 mRNA expression in the liver was quantified through RT–PCR. The pro-inflammatory cytokines, TNF-α (**D**), IL-6 (**E**), and IL-1β (**F**), in the serum were quantified through enzyme-linked immunosorbent assays. Bar graph values indicate the mean ± standard deviation (*n* = 3−5 rats per group per time-point). † *p* < 0.05 and †† *p* < 0.01 versus negative control rats (OLP 0 mg/kg, time 0 h). * *p* < 0.05 and ** *p* < 0.01 versus PH/LPS-only rats.

**Figure 3 ijms-25-07189-f003:**
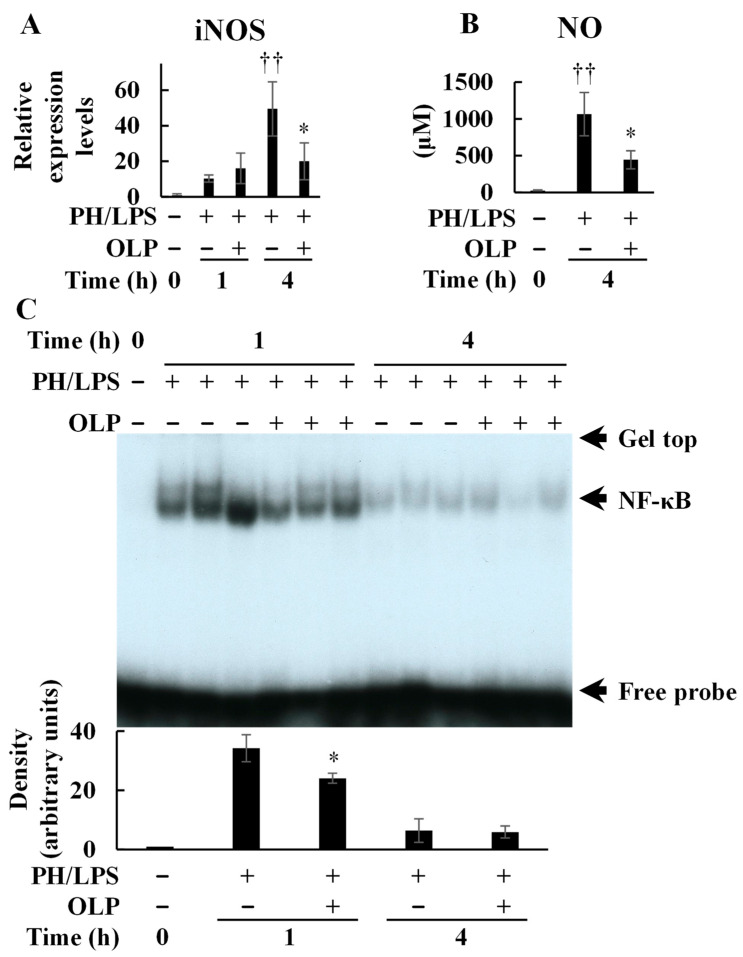
Suppression of nitric oxide production and NF-κB activation by olprinone (OLP). Effects of OLP on NO production and NF-κB activation in PH/LPS-treated rats. (**A**) iNOS mRNA expression in the liver was analyzed using RT-qPCR. (**B**) Serum NO levels. (**C**) NF-κB activation was analyzed by an electrophoretic mobility shift assay (upper). The band densities were measured (lower). Bar graph values indicate the mean ± standard deviation (*n* = 1−5 rats per group per time-point). †† *p* < 0.01 versus normal negative control rats (OLP 0 mg/kg, time 0 h). * *p* < 0.05, versus PH/LPS-only rats.

**Figure 4 ijms-25-07189-f004:**
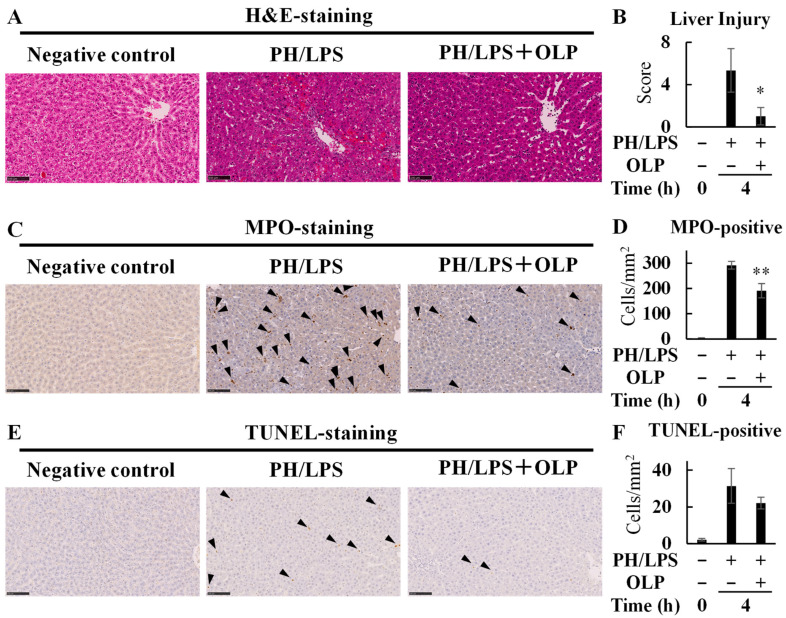
Alleviation of PH/LPS-induced histopathological damage by olprinone (OLP). (**A**) Histological findings of the PH/LPS-treated liver. Liver samples were obtained from rats in the untreated group (left) and from rats in the PH/LPS-only group (middle) and the PH/LPS + OLP group (right) 4 h after administration of LPS. The sections were stained with hematoxylin and eosin (H&E) (magnification ×200, bar = 100 μm). (**B**) Liver injury in (**A**) was scored. Effects of OLP on neutrophil infiltration (**C**,**D**) and apoptosis (**E**,**F**) in the liver. (**C**) The sections were immunostained for myeloperoxidase (MPO; arrowhead: positive cell, magnification ×200, bar = 100 μm). (**D**) The numbers of positive cells were counted per square millimeter. (**E**) Liver sections were stained through TUNEL (magnification ×200, bar = 100 μm, arrowhead: positive nucleus). (**F**) The numbers of positive nuclei were counted per square millimeter. Bar graph values indicate the mean ± standard deviation (*n* = 3 rats per group per time-point). * *p* < 0.05 and ** *p* < 0.01 versus PH/LPS-only rats.

**Figure 5 ijms-25-07189-f005:**
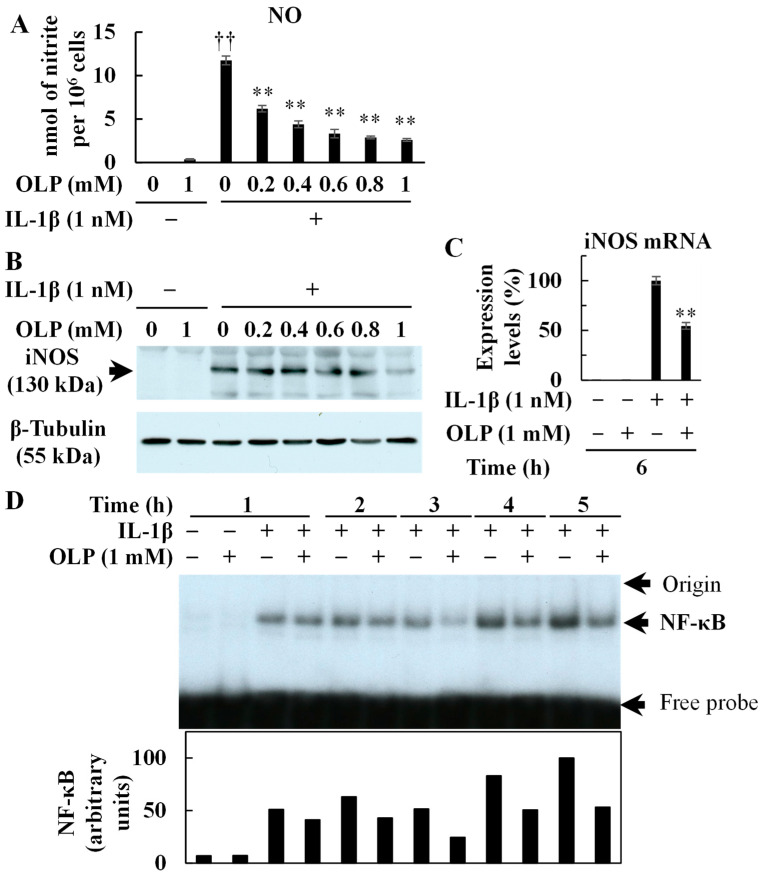
Suppression of NO production and NF-κB activation in primary cultured rat hepatocytes by olprinone (OLP). (**A**) Nitrite concentration was measured in the culture medium of the cells treated with IL-1β and OLP for 8 h. (**B**) The expression of iNOS (upper) and β-tubulin protein (internal control; lower) were analyzed through Western blotting. Cells were treated with IL-1β and OLP for 8 h. (**C**) The iNOS mRNA expression was quantified with RT-PCR in IL-1β–stimulated hepatocytes. Bar graph values indicate the mean ± standard deviation (*n* = 3). †† *p* < 0.01 versus untreated control (IL-1β 0 nM, OLP 0 mM). ** *p* < 0.01 versus IL-1β only. (**D**) NF-κB activation after IL-1β treatment was analyzed through electrophoretic mobility shift assay (upper). The band densities were measured (lower).

**Figure 6 ijms-25-07189-f006:**
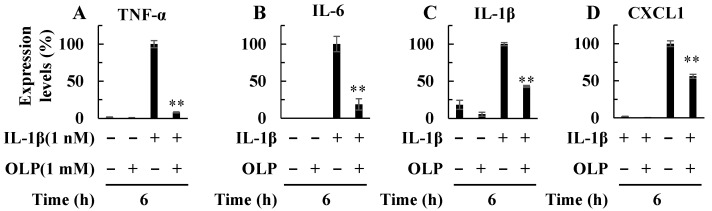
Suppression of mRNA expression of inflammatory mediators by olprinone (OLP). The mRNA expression of TNF-α (**A**), IL-6 (**B**), IL-1β (**C**), and CXCL1 (**D**) was quantified through RT-PCR. Cells were treated with IL-1β and OLP. Bar graph values indicate the mean ± standard deviation (*n* = 3). ** *p* < 0.01 versus IL-1β only.

## Data Availability

Data is contained within the article and Appendix A.

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
