# Peer review of "Olprinone, a Selective Phosphodiesterase III Inhibitor, Has Protective Effects in a Septic Rat Model after Partial Hepatectomy and Primary Rat Hepatocyte"

_ijms, 2024, doi:10.3390/ijms25137189_

Round 1

Reviewer 1 Report

Comments and Suggestions for Authors

This paper by Kotsuka et al. is interesting and it addresses the sepsis caused by postoperative infection after hepatic resection that are still important challenges in hepatic surgery procedures. The fact that Olprinone (OLP) can be used to improve the survival rate in postoperative infection conditions by reducing proinflamamtory responses, both mRNA expression and molecules/cytokines release, liver injury and apoptosis.

However, proinflammatory reaction are required also to promote the regeneration of the damaged/injured organ (in this case the liver). Survival rate up to 5 days is drastically improved, but no data on the liver structure are shown other than 4 hours. Does the treatment with OLP delay or accellerate the liver regeneration? It would be interesting and it would complete the paper showing that there are no delays in liver regeneration or that the regeneration is improved by OLP treatment

OLP treatment is performed 1 hour before LPS injection. What would be the outcome if the OLP is given immediately before/during/immediately after surgery (48 hours before LPS injection)? Would it be possible to envisage a post-operative treatment with OLP along with antibiotics, thus about 1 week at a lower dosage every day? Or would it give accumulation/overdosage problems?

Author Response

ijms-2989946, entitled “Olprinone, a selective phosphodiesterase â…¢ inhibitor, has protective effects in a septic rat model after partial hepatectomy and primary rat hepatocyte”

First of all, we thank the reviewer for their invaluable comments and suggestions. We have addressed all the comments and revised the text. Please find our responses to the comments below.

This paper by Kotsuka et al. is interesting and it addresses the sepsis caused by postoperative infection after hepatic resection that are still important challenges in hepatic surgery procedures. The fact that Olprinone (OLP) can be used to improve the survival rate in postoperative infection conditions by reducing proinflamamtory responses, both mRNA expression and molecules/cytokines release, liver injury and apoptosis.

However, proinflammatory reaction are required also to promote the regeneration of the damaged/injured organ (in this case the liver). Survival rate up to 5 days is drastically improved, but no data on the liver structure are shown other than 4 hours. Does the treatment with OLP delay or accellerate the liver regeneration? It would be interesting and it would complete the paper showing that there are no delays in liver regeneration or that the regeneration is improved by OLP treatment

- In this study, the effects on liver injury and inflammatory response were examined using a lethal model in which more than half of the animals die at 24 hours. Under these conditions, it was difficult to sample longer than 4 hours, especially from 24 to 48 hours, when liver regeneration is maximally accelerated.

As the reviewer pointed out, proinflammatory reactions promote liver regeneration; it would be worthwhile to clarify whether OLP promotes liver regeneration and how proinflammation is involved in this process. Investigating the relationship between anti-inflammatory effects and liver regeneration when treated with OLP in models of milder injury is an important future study. These are added to the discussion.

Line 214

On the other hand, pro-inflammatory reactions are important for liver regeneration. Investigating the association between anti-inflammatory effects and liver regeneration when treated with OLP in models of milder injury is an important topic for future studies.

OLP treatment is performed 1 hour before LPS injection. What would be the outcome if the OLP is given immediately before/during/immediately after surgery (48 hours before LPS injection)? Would it be possible to envisage a post-operative treatment with OLP along with antibiotics, thus about 1 week at a lower dosage every day? Or would it give accumulation/overdosage problems?

-The hepatoprotective effect of continuous administration of OLP in the swine partial hepatectomy model has been reported. Liver injury is alleviated by preoperative administration but aggravated by both preoperative and postoperative administration (Ref.1).

In the present model of septic hepatotoxicity, only the preoperative dose was administered. The therapeutic effect of postoperative administration, including continuous administration, is a subject for future study. As the reviewer pointed out, the effect of accumulation/overdosage on therapeutic outcomes is also a major issue. These are added to the discussion.

Ref.1 Iguchi K.; Hatano E.; Yamanaka K.; Sato M.; Yamamoto G.; Kasai Y.; Okamoto T.; Okuno M.; Taura K.; Fukumoto K.; Ueno K.; Uemoto S. Hepatoprotective effect by pretreatment with olprinone in a swine partial hepatectomy model. Liver Transpl 2014 20, 838-849. DOI: 10.1002/lt.23884.

Line 227

The impact of cumulative/overdosage on therapeutic outcomes is another important issue that should be investigated in the future.

Reviewer 2 Report

Comments and Suggestions for Authors

This is an interesting study by Kotsuka and colleagues. Liver injury and sepsis following extensive liver resection is a devastating complication and currently there are no treatment options for post hepatectomy liver failure. The authors are to be congratulated on a well designed and well written study.

The experimental methodology is sound but I wonder if the authors could clarify 2 points.

1: The have shown a reduction in liver injury score as measured by histology in rats who were treated with Olprinone however this is not reflected in a reduction in AST or ALT which is conflicting as ALT and AST are good markers for hepatocyte damage and necrosis.

2: In figure 3B they have only shown NO levels at 4 hours. All other markers were measured and presented at 1 and 4 hours. Can the authors explain why the NO levels at 1 hour are not presented?

Otherwise this is a well written manuscript and an interesting body of work.

Author Response

ijms-2989946, entitled “Olprinone, a selective phosphodiesterase â…¢ inhibitor, has protective effects in a septic rat model after partial hepatectomy and primary rat hepatocyte”

First of all, we thank the reviewer for their invaluable comments and suggestions. We have addressed all the comments and revised the text. Please find our responses to the comments below.

This is an interesting study by Kotsuka and colleagues. Liver injury and sepsis following extensive liver resection is a devastating complication and currently there are no treatment options for post hepatectomy liver failure. The authors are to be congratulated on a well designed and well written study.

The experimental methodology is sound but I wonder if the authors could clarify 2 points.

1: The have shown a reduction in liver injury score as measured by histology in rats who were treated with Olprinone however this is not reflected in a reduction in AST or ALT which is conflicting as ALT and AST are good markers for hepatocyte damage and necrosis.

-As shown in Figure 1, ALT and AST levels varied widely among individuals and were not significantly induced, or found to be reduced by OLP treatment; AST was relatively inducible and showed a trend toward reduction by OLP at 4 hours. As described by the reviewers, ALT and AST are good markers of hepatocyte damage and necrosis. Therefore, these results are presented in the manuscript.

On the other hand, since AST is also found in cardiac muscle, skeletal muscle, and red blood cells, there may have been an effect on damage in these tissues. Obviously, it is important to obtain reproducible results, but investigating damage to tissues other than the liver and the effect of OLP treatment in animal models is a topic for future work.

2: In figure 3B they have only shown NO levels at 4 hours. All other markers were measured and presented at 1 and 4 hours. Can the authors explain why the NO levels at 1 hour are not presented?

-In a previous report, NO levels became more pronounced after 3 hours in the PH/LPS model, and the decrease in NO levels was significant in the drug study (Ref. 1). Therefore, customarily, NO levels are measured only at 4 hours (Ref. 2). New references and explanations were added.

Ref.1 Tsuchiya, H.; Kaibori, M.; Yanagida, H.; Yokoigawa, N.; Kwon, A. H.; Okumura, T.; Kamiyama, Y. Pirfenidone prevents endotoxin-induced liver injury after partial hepatectomy in rats. J Hepatol 2004 40, 94–101, DOI:10.1016/j.jhep.2003.09.023.

Ref.2 Nakatake, R.; Okuyama, T.; Kotsuka, M.; Ishizaki, M.; Kitade, H.; Yoshizawa, K.; Tolba, R.H.; Nishizawa, M.; Sekimoto, M. Combination Therapy with a Sense Oligonucleotide to Inducible Nitric Oxide Synthase mRNA and Human Soluble Thrombomodulin Improves Survival of Sepsis Model Rats after Partial Hepatectomy. Shock 2023, 60, 84-91, DOI:10.1097/SHK.0000000000002135.

Line 101

Since serum NO levels are more pronounced after 3 h of PH/LPS treatment [14], NO levels were measured at 4 h in this study and showed a significant increase.

Round 2

Reviewer 1 Report

Comments and Suggestions for Authors

The authors have answered all the points. No additional comments

Author Response

We thank the reviewer.